# Influence of Confining Pressure on Nonlinear Failure Characteristics of Coal Subjected to Triaxial Compression

**DOI:** 10.3390/ijerph20010105

**Published:** 2022-12-21

**Authors:** Qiuping Li, Jie Liu, Shouqing Lu, Zaiquan Wang, Hao Wang, Yimeng Wu, Yupu Wang, Di Ying, Mingjie Li

**Affiliations:** 1Department of Safety Engineering, Qingdao University of Technology, Qingdao 266520, China; 2Key Lab of Industrial Fluid Energy Conservation and Pollution Control, Qingdao University of Technology, Ministry of Education, Qingdao 266520, China; 3Shandong Key Industry Field Accident Prevention Technology Research Center (Non-Ferrous Metallurgy), Qingdao 266520, China; 4School of Civil Engineering, Qingdao University of Technology, Qingdao 266520, China

**Keywords:** triaxial compression, crack propagation, damage variable, coal-body destruction

## Abstract

The stress of a coal seam increases with an increase in the mining depth, which makes the failure mechanism of a coal mass more complex. To reveal the deformation and failure law of deep coal, a series of triaxial experiments was carried out via laboratory experiments and numerical simulation experiments to analyze the influence of the confining stress on the nonlinear failure characteristics of coal. Based on the crack-propagation model, the values for the inelastic flexibility *S*_1_ and the damage variable *D* were calculated. The results showed that the value of *S*_1_ decreased with an increase in the confining stress, which indicated that the increase in the confining pressure could inhibit the crack propagation and that the inhibitory effect was more obvious when the confining pressure increased in a small range of 4 to 12 MPa. The damage variable decreased with an increase in the confining pressure at the yield point; moreover, with an increase in the initial confining pressure, the damage rate gradually decreased. The coal body changed from the compression state to the expansion state when moving from the yield point to the peak point, and the compression value of the yield point and the dilation value of the peak point increased with the increase in the confining pressure. After the coal body entered the yield stage, the change in the confining pressure had a more significant effect on the damage to the coal body.

## 1. Introduction

In recent years, coal mining has been developing toward large-scale and deep mining. The stress concentration caused by engineering excavation is obvious, and the ground stress is increasing. The high-risk attributes of coal mining operations are still very significant [1]. In addition, there are a large number of primary defects such as pores and microcracks in coal, and the process of deformation and failure under load is very complex. The stress of a coal seam increases significantly with an increase in the excavation depth, and the deformation and failure characteristics of the deep coal rock are obviously different from those of shallow coal rock [2,3]. Fractures in natural rocks play an important role in determining the strength, deformability, and failure behavior of rock masses; to increase the strength of fractured rock masses, rock-reinforcement methods are always used in underground mining [4,5,6,7,8,9]. Frequent coal mining accidents seriously threaten people’s lives and property safety [10,11,12,13,14]. Therefore, a comprehensive analysis of the effect of the confining stress on the deformation and failure characteristics of coal and rock can help to better understand the complex deformation and failure process of deep coal rock.

With an aim to reveal the influence of the confining pressure on the deformation and failure characteristics of coal and rock, in recent years many scholars have carried out mechanical tests under different confining pressures and analyzed the variations in the strength and deformation of coal and rock masses caused by the confining pressure. Liu et al. [15] studied conventional triaxial tests of coal and rock under different confining pressures and concluded that the elastic modulus and peak strength of the coal and rock increased with an increase in the confining pressure; the higher the confining pressure, the more obvious the ductility characteristics of the coal and rock. Xu et al. [16] carried out triaxial compression test on rock, analyzed the dilatancy law of rock in the process of deformation and failure, and found that the stress of the characteristic point of the rock increased with an increase in the confining pressure, its dilatancy characteristics decreased, and the dilatancy index had a confining pressure effect. Zhang et al. [17] tested the deformation, failure, and acoustic emission characteristics of gas-bearing coal under triaxial loading and discussed the influence of the gas pressure and confining stress on the deformation, failure, and fractal characteristics of the coal.

The propagation process of an initial crack plays an important role in the deformation and failure of a coal–rock mass. To further reveal the effects of a crack on the deformation and failure of a coal–rock mass, some scholars carried out experimental research. Zhu et al. [18] conducted conventional triaxial compression experiments on granite, studied the change law of the crack strain, and obtained the crack-initiation stress of granite under different confining pressures. The study found that as the confining pressure increased, the original crack closure of the granite decreased and the elastic stage experienced was longer. After entering the crack-propagation stage, the crack-propagation speed became slower. Yang et al. [19,20] carried out traditional triaxial compression experiments to study the strength, deformation ability, and failure behavior of coal–rock specimens with cracks. Li et al. [21] established an axial stress–strain constitutive model that considered the influence of the crack angle and explained the constitutive relationship between the confining pressure and axial strain. In addition, many scholars [22,23] considered the influence of the confining pressure on the initiation angle of wing cracks and analyzed the influence of friction between closed crack walls [24], the confining pressure, and the microcrack length on the progressive failure process of rock under crack propagation based on the evolution law of the crack-propagation intensity factor in fracture mechanics [25].

Under the action of an external force, the microdefects distributed inside a coal–rock mass will continue to aggregate, expand, and penetrate, thereby resulting in the damage evolution and failure mode of the internal materials of the coal–rock mass. Some scholars have studied the damage of coal–rock materials. Zhang et al. [26] deduced the calculation formula of the elastic-plastic damage variable of coal–rock under cyclic loading and analyzed the evolution behavior of the elastic-plastic damage. Xu [27] carried out triaxial compression tests of high-stress dacite under different confining pressures and compared the damage-accumulation process of the rock under high stress. It was concluded that the damage accumulation of a low confining pressure was faster and that the confining pressure may inhibit the cumulative expansion of rock damage. Chen et al. [28,29] used UDEC to analyze the damage process of loaded brittle rocks and found that reasonable stress–strain behavior and microscopic and macroscopic mechanical damage evolution could be reproduced by the proposed DEM model. In addition to laboratory tests, some scholars [30,31] have used numerical simulations to gain a deeper understanding of the deformation and damage behavior of rock samples under triaxial loading.

At present, scholars have carried out a large number of triaxial compression tests and established constitutive models from the perspective of crack initiation, propagation and damage, which has enriched the study of the confining-pressure effect on the mechanical properties of rock. However, most of the tests took rock and sandstone as the research objects and seldom considered the inhibition effect of the confining pressure on microcracks in the process of triaxial compression of coal and rarely systematically compared and analyzed the crack propagation and damage accumulation. Therefore, numerical simulation and laboratory testing of the triaxial compression of coal under different confining pressures were carried out in the present study. While considering the influence of the confining stress on the propagation of microcracks in coal, the damage-accumulation process of coal was analyzed, and the entire process of the deformation and failure of coal was systematically analyzed.

## 2. Experiment Analysis

### 2.1. Experimental System and Process

Each coal sample was processed into a standard cylindrical sample with dimensions of 50 mm × 100 mm, and the triaxial compression test was completed by using a rock multi-field coupling triaxial tester. The test system is shown in Figure 1. The system could be used to study the deformation and failure of rock materials under a high confining pressure, which consisted of the confining pressure and the axial pressure:

(1) Confining pressure loading: this part could meet the two working needs of pressure and flow, and the pressure and loading and unloading speeds during the test could be adjusted via the cylinder system.

(2) Axial compression loading: This part could realize stress loading and unloading through pressure and axial and lateral displacement control through a computer-controlled program to realize the accurate control of the stress state of the rock material.

**Figure 1 ijerph-20-00105-f001:**
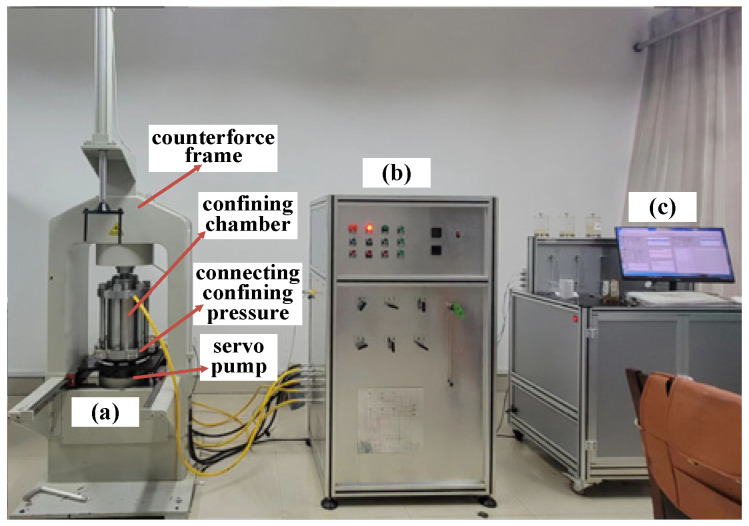
Experimental system: (**a**) principal pressure testing machine; (**b**) hydraulic power supply; (**c**) software control system.

With an increase in the excavation depth in coal mining, the stress state of the coal and rock becomes more and more complex. In this paper, to better study the mechanical properties of coal and rock at different depths and cause the change in the confining pressure to make a significant difference in the deformation and failure of the coal while considering the loading capacity of the test equipment, triaxial tests with confining pressures of 4 MPa, 12 MPa, 20 MPa, and 28 MPa were respectively carried out. The specific test process was as follows. Firstly, according to the hydrostatic pressure condition, the force-control method was used to cause the axial stress *σ*_1_, radial stress *σ*_2_, and *σ*_3_ of the specimen to satisfy *σ*_1_ = *σ*_2_ = *σ*_3_ = 4, 12, 20, 28 MPa at 500 N/s. Then, the confining stress was stabilized, and the displacement-control method was used to increase the axial pressure gradually at a displacement rate of 0.05 mm/s until the coal sample was destroyed. The stress path is shown in Figure 2.

### 2.2. Experimental Results

#### 2.2.1. Effect of Confining Pressure on Stress and Strain

Figure 3 shows the stress–strain curves under different confining pressures in the laboratory tests. In Figure 3, it can be seen that the change trend of the stress–strain curves under different confining pressures was basically the same and could be divided into four stages:

(1) Compaction stage: the microcracks in the coal body were closed under the action of external force loading and there was a short compaction stage. The speed of the strain increase became larger, and the curve showed a ‘downward convex’ type.

(2) Elastic stage: the stress under different confining pressure increased linearly with the strain. Due to the increase in the confining pressure, the strength limit of the coal body became larger so that the elastic stage of the coal body became longer with the increase in the confining pressure.

(3) Yield stage: because the increase in the confining pressure limited the radial deformation of the coal, the energy stored in the coal was consumed in the form of dissipated energy. Therefore, with the increase in the confining pressure, the yield stage of the coal became longer.

(4) Post-peak stage: the test results showed that when the confining pressure was 4 MPa, the brittleness characteristics of the coal body after the peak were obvious. With the increase in the confining pressure, the coal body showed more obvious ductility characteristics. Under the confining pressure of 28 MPa, the stress-change curve of the coal body after the peak was almost a plastic flow state.

**Figure 3 ijerph-20-00105-f003:**
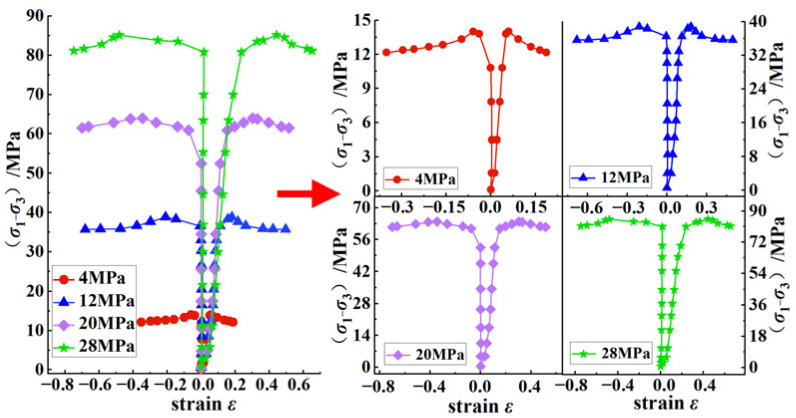
Stress–strain curves under different confining pressures in laboratory tests.

The above phenomenon showed that although the deformation and failure stages of the coal under different confining pressures were similar, there were obvious differences in the changes in mechanical parameters such as the strength and strain of the coal bodies under different confining pressures, which indicated that the deformation and failure process of the coal was greatly affected by the confining pressure. Therefore, it was necessary to study the influence of the confining pressure on the entire process of coal failure in this study.

The stress and strain of the yield point and peak point are very important when judging the degree and speed of a coal fracture. The yield point is the point at which the coal body ends the elastic stage and begins to undergo plastic deformation; the peak point is the highest point on the stress–strain curve of the coal body (that is, the point that corresponds to the maximum stress of the coal body). Figure 4 shows the stress and strain curves that corresponded to the yield point and the peak point under different confining pressures; it can be seen that at the yield point of the coal, the axial strain was greater than the radial strain under each confining pressure. At the peak point, the radial strain under each confining pressure was greater than the axial strain, and the difference between the axial strain and the radial strain at the yield point and the peak point both increased with the increase in the confining pressure. The above phenomenon showed that the coal was compressed at the yield point and that the compression phenomenon of the coal under a high confining pressure was more obvious. The coal was in a state of expansion at the peak point, and the phenomenon under a high confining pressure was more obvious.

During the loading process from the yield point to the peak point, the coal under each confining pressure changed from the compression state to the expansion state; this process could be divided into two stages: Stage I included the yield point to the expansion point (the point at which the volume strain appeared as negative); Stage II included the expansion point to the peak point. In Stage I, the time when the coal body changed from the maximum compression state to the expansion state increased with the increase in the confining pressure; the times were 5 s, 11 s, 17 s, and 23 s, respectively. This indicated that the increase in the confining pressure could delay the time when the crack entered the accelerated expansion stage to a certain extent while the change rate of the volumetric strain gradually increased with the increase in the confining pressure as shown in Figure 5. In Stage II, the coal body was in the expansion state, and the volume-expansion rate of the coal body increased with the increase in the confining pressure as shown in Figure 5. This showed that after the coal entered the damage stage, the energy accumulated by the larger initial confining pressure before the damage had a significant effect on the coal damage after the coal entered the yield stage, thereby resulting in the coal moving from the compression state to the expansion state at a faster rate.

#### 2.2.2. Effect of Confining Pressure on Deformation Parameters

Coal is a nonlinear elastic material with a deformation modulus that changes with stress–strain during loading, so the secant modulus (calculated from two data points close to each other in the axial-stress and axial-strain curve) is used for a coal body in the elastic-deformation stage [28]:(1)$$E_{T}=\frac{\varepsilon_{i+1}-\varepsilon_{i}}{\sigma_{i+1}-\sigma_{i}}$$
where *E_T_* is the secant modulus in the elastic-deformation stage, *ε_i_*_+1_ and *ε_i_* are the axial strain at each moment, and *σ_i_*_+1_ and *σ_i_* are the axial stress at each moment.

After the coal enters the yield and failure stage, the influence of the stress and strain in different directions should be considered when calculating the deformation parameters under triaxial compression. According to the generalized Hooke’s law, it is assumed that in the process of coal failure caused by triaxial compression, each stress and deformation point on the stress–strain curve of the sample conforms to the generalized Hooke’s law. This can be obtained as [32,33]:(2)$$\left\{ \begin{array}{l} E_{C}=(\sigma_{1}-2v\sigma_{3})/\varepsilon_{1}\\ v_{c}=(B\sigma_{1}-\sigma_{3})/[\sigma_{3}(2B-1)-\sigma_{1}]\\ B=\varepsilon_{3}/\varepsilon_{1} \end{array}\right.$$
where *E_C_* is the deformation modulus, *v_c_* is transverse-deformation ratio, *B* is the expansion coefficient in the plastic-deformation stage, *ε*_1_ is the axial strain, and *ε*_3_ is the radial strain.

Figure 6 shows the change trend of the deformation modulus *E* and the transverse deformation ratio *v* with axial strain under different confining pressures in the entire process of coal deformation and failure. It can be seen that the deformation modulus and transverse-deformation ratio decreased slightly with the increase in the axial strain and then gradually returned to the initial value. Finally, after reaching a critical value, the deformation modulus decreased greatly and the transverse-deformation ratio increased greatly. Since the microcracks inside the coal body were closed under the action of stress during the compaction stage, the axial strain increased rapidly at this stage, so the deformation modulus and transverse-deformation ratio both decreased slightly. At the same time, the deformation modulus under a high confining pressure was greater than that under a low confining pressure, which indicated that the increase in the confining pressure could improve the ability of the coal to resist deformation. The transverse-deformation ratio under a high confining pressure increased rapidly when the axial strain was large; under the same axial strain condition, the transverse-deformation ratio decreased with the increase in the confining pressure, which indicated that when the coal body was loaded by an external force, the radial strain value of the coal body decreased with the increase in the confining pressure, thus indicating that the increase in the confining pressure under certain conditions could improve the resistance of the coal body to lateral deformation.

## 3. Numerical Experiment

### 3.1. Numerical Simulation Program

The sample size of the numerical simulation was the same as that of the laboratory test (a 50 mm × 100 mm standard cylinder). In this simulation, a two-dimensional axisymmetric model was selected; the boundary conditions of this model are shown in Figure 7 The sample parameters and stress paths simulated in this paper were consistent with the laboratory test; the specific material parameters are shown in Table 1.

### 3.2. Comparison of Experimental and Numerical Simulation Results

Figure 8 shows the stress–strain curves of the simulation tests and the experiments under different confining pressure conditions. It can be seen that the simulation test under different confining pressures was basically consistent with the experiments. The axial strain *ɛ*_1_ of the simulation test increased linearly with the stress difference (*σ*_1_ − *σ*_3_) in the initial loading stage, and there was no obvious compaction stage. The reason for this phenomenon was that the distribution of particles inside the coal body was relatively uniform during the generation of the numerical model, and the internal defects were not obvious. However, there were primary microcracks in the coal samples in the laboratory test, and the distribution was discrete. Therefore, the stress–strain curve of the laboratory test had an obvious compaction stage, which led to the incomplete coincidence of the curves of the simulation test and of the laboratory test under different confining pressures.

### 3.3. Coal Failure Process

Figure 9 shows the change in the plastic deformation of the coal under different confining pressures with the loading step. It can be seen that when the initial confining pressure was 4 MPa, 12 MPa, 20 MPa, and 28 MPa, the corresponding loading steps when the plastic deformation of the coal began to appear were the 56th step, the 152nd step, the 251st step, and the 350th step, respectively, which indicated that the increase in the confining pressure could improve the bearing capacity of the coal body.

When the coal body began to show plastic deformation, the plastic-deformation area in the coal body increased significantly under a low confining pressure with the increase in the loading step; under a high confining pressure, the failure area in the coal expanded slowly with the increase in the loading step and the plastic deformation area was small. When the confining pressure was loaded to the 350th step, it can be seen in Figure 9a,b that the coal bodies under 4 MPa and 12 MPa were both in the post-peak failure stage and were completely destroyed. When the confining pressure was 20 MPa, the coal body was in the yield stage, and the plastic-deformation area of the coal body was large but it was not completely destroyed; when the confining pressure was 28 MPa, the coal began to show plastic deformation in the 350th step, and the deformation area was very small.

The above coal failure phenomenon showed that the failure time of the coal body was late under a high confining pressure. Before the yield stage, the failure area and the expansion speed of the coal body decreased with the increase in the confining pressure, which indicated that the damage to the coal body had a time effect, and the increase in the confining pressure could delay the failure time of coal body.

## 4. Discussion

### 4.1. Effect of Confining Pressure on Crack Propagation

The deformation and failure of a coal body is a process of the extension of internal cracks; the confining pressure changes the mechanical properties of the coal body by affecting the propagation law of internal cracks in the body. Therefore, analyzing the influence of the confining pressure on crack initiation and extension is helpful to deepen the understanding of the influence mechanism of the confining pressure on the mechanical properties of a coal body.

Zuo et al. [34] derived an axial strain model for pre-peak cracking based on the definition of natural strain. According to the definition of natural strain, the crack strain after damage intensity is:(3)$$d\varepsilon_{1}^{c}=\frac{dh_{c}}{h_{c}}$$
where $\varepsilon_{1}^{c}$ is the axial strain of the crack, and *h*_c_ is the axial equivalent height of the crack-propagation stage. The stress difference on the surface of coal (including cracks and matrix) is:(4)$$d(\sigma_{1}-\sigma_{3})=\frac{d\varepsilon_{1}^{c}}{S_{1}}$$
where *S*_1_ is the inelastic flexibility of the crack (which characterizes the deformation degree of the coal). Equation (3) can be substituted into Equation (4) and integrated to get:(5)$$\sigma_{1}-\sigma_{3}=\frac{\ln h_{c}}{S_{1}}+C$$

Before a crack in coal begins to expand, the crack-initiation strain is very small and is difficult to be measured by the test. Therefore, while assuming that *H*_c_ is the axial equivalent height when the crack begins to expand, the initial condition of crack propagation can be obtained. When *σ*_1_ − *σ*_3_ = *σ*_s_ then *H*_c_ = *h*_c_, and the integral constant can be obtained by using the initial condition into Equation (5):(6)$$C=\sigma_{S}-\frac{\ln H_{C}}{S_{1}}$$

The axial displacement of the crack Δ can be obtained by substituting Equation (5) into Equation (6):(7)$$\Delta =h_{c}-H_{c}=H_{c}\{\exp[S_{1}(\sigma_{1}-\sigma_{3}-\sigma_{S})-1]\}$$

The relationship between the axial strain and axial stress of the crack can be obtained from Equation (7):(8)$$\varepsilon_{1}^{c}=\frac{H_{c}}{H}\{\exp[S_{1}(\sigma_{1}-\sigma_{3}-\sigma_{S})-1]\}$$
where *H* is the height of the specimen. Let $\varepsilon_{1}^{c0}$ be the axial strain of initial crack propagation; its value can be expressed as $\varepsilon_{1}^{c0}=H_{c}/H$. Then, Equation (8) can be transformed into:(9)$$\varepsilon_{1}^{c}=\varepsilon_{1}^{c0}\{\exp[S_{1}(\sigma_{1}-\sigma_{3}-\sigma_{S})-1]\}$$

Based on the simulation test and laboratory test data, the crack-propagation model was respectively fitted. The axial stress–strain-difference test data and fitting curve of the crack under different confining pressures are shown in Figure 10; it can be seen that the data obtained from the simulation and the indoor test under different confining-pressure conditions fit well with the model. The correlation coefficient *R*^2^ of the model was above 0.97; therefore, the fitting parameters had a good reference effect.

In the process of coal mine excavation, the strain of a coal body can be decomposed into two parts: one is the crack strain caused by the closure, initiation, expansion, and penetration of cracks in the coal body during the loading process; the other is the elastic strain caused by the elastic deformation of the coal matrix. To quantitatively analyze the crack evolution in the process of rock failure, this paper adopted the following calculation formula for the crack axial strain under triaxial compression [32]:(10)$$\varepsilon_{1}^{c}=\varepsilon_{1}-\frac{1}{E_{S}}(\sigma_{1}-2v\sigma_{3})$$
where *ɛ*_1_ is the axial strain of the coal; $\varepsilon_{1}^{c}$ is the axial strain of the crack; and *E*_s_ and *v* are the elastic modulus and Poisson’s ratio, respectively, obtained from the test curve in the elastic stage.

The mechanical analysis data in this section were based on numerical simulation results. Figure 11 shows the axial stress–strain curves of cracks under different confining pressures as calculated using Equation (10). It can be seen in Figure 11 that the axial stress–strain curves of the cracks under different confining pressures could be divided into three stages: the elastic stage, the crack-propagation stage, and the post-peak stage. Firstly, with the increase in the stress difference, the axial strain of the crack was almost 0, which indicated that the coal body mainly underwent elastic deformation at this stage; that is, the coal body was in the elastic stage. Secondly, when the stress difference increased to the yield stress *σ*_s_, the axial strain of the crack began to increase significantly, which indicated that the crack entered the rapid-expansion stage after the yield stress. Finally, when the stress difference continued to increase to the peak stress *σ*_p_, the bearing capacity of the coal decreased rapidly; in the meantime, the axial strain of the crack increased greatly, and the coal entered the post-peak stage.

Figure 12 shows the curves of $\varepsilon_{1}^{c0}$, *σ*_s_, and *S*_1_ with confining pressures obtained using Equation (9); it can be seen that the yield strength *σ*_s_ of the coal increased linearly with the increase in the confining pressure. The axial crack strain $\varepsilon_{1}^{c0}$ and inelastic flexibility *S*_1_ both decreased with the increase in the confining stress, but it can be seen in Figure 10 that the axial crack strains $\varepsilon_{1}^{c0}$ of the crack propagation under different confining pressures were both very small: the value ranges were 8.2 × 10^−6^~1.27 × 10^−4^ and 7.61 × 10^−6^~1.26 × 10^−4^, respectively. When combining the above phenomenon with the data shown in Figure 10, it can be seen that in Stage II of the crack propagation, the larger the confining pressure, the greater the axial strain change in the crack, which indicated that the increase in the confining pressure made the yield stage of the coal body longer, the deformation that the coal body could withstand in this stage was increased, and the coal body had a greater ability to resist damage.

### 4.2. Effect of Confining Pressure on Damage Variable

The changes in the deformation modulus with the stress difference calculated using Equations (1) and (2) in Section 2.2.2 are shown in Figure 13. The variation trend conformed to the exponential function, and the deformation modulus tended to be stable within a certain range of stress difference. Finally, it suddenly decreased when the stress difference reached a certain value. The sudden decrease in the deformation modulus of the coal body reflected the cumulative damage in the sample [35]. Therefore, the change formula of the deformation modulus and stress is simplified as:(11)$$E=a-k\exp(\sigma_{1}-\sigma_{3}-b)$$
where *a*, *k*, and *b* are the fitting parameters. The deformation modulus in the process of coal deformation failure can be expressed as [36]:(12)$$E=E_{0}(1-D)$$
where *E*_0_ is the initial deformation modulus.

By observing the change in the deformation modulus, when the stress difference was very small, *a* was close to 450 MPa, which was equal to the initial elastic modulus of the coal, so it could be regarded as *a* = *E*_0_; then the damage variable formula could be obtained from Equations (11) and (12):(13)$$D=\frac{k}{a}\exp(\sigma_{1}-\sigma_{3}-b)$$
where the range of damage variable *D* is 0~1. When *D* = 0, it meant that the coal body was not damaged; when *D* = 1, it meant that the coal body was completely damaged. The damage variable *D*_s_ = (*k*/*a*) exp (*σ*_s_ − *b*) corresponding to the yield stress *σ*_s_ and the damage variable *D*_c_ = (*k*/*a*) exp (*σ*_c_ − *b*) corresponding to peak stress *σ*_c_ could be determined using Equation (13).

Figure 13 shows the fitting results of the data of the deformation modulus changing with the stress difference as calculated using Equation (11). It can be seen in Figure 13 that the data fit well in the pre-peak stage and that the value of the damage variable in the pre-peak stage could be calculated from the fitting results. The fitted parameters and the calculated damage variables at the yield point and peak point are shown in Table 2.

Figure 14 shows the variations in the stress difference, damage variable, and crack axial strain with the axial strain of the coal under different confining pressures. It can be seen in Figure 14 that the accumulation process of the damage variable was related to the crack-propagation stage: in the early elastic stage, the crack was in the compaction stage, the axial strain of the crack was 0, and the damage variable *D* = 0; in the later stage of the elastic stage, the crack began to initiate with the increase in stress, the axial strain of the crack increased with a very small amplitude, and the increased amplitude decreased with the increase in the confining pressure. At this time, the damage variable began to accumulate gradually. The damage variable increased slightly under the confining pressures of 4 MPa and 12 MPa, while the increase was almost 0 at 20 MPa and 28 MPa. When the coal body entered the yield stage, the crack entered the rapid-expansion stage. At this time, the damage variable increased rapidly. The above phenomenon indicated that the confining pressure could play a role in reducing the damage degree of the coal body in the elastic stage and the early stage of the yield stage and that the increase in the confining pressure could improve the ability of the coal body to withstand damage.

During the stage of stable crack propagation, the variations in the axial strain of the crack under confining pressures of 4, 12, 20, and 28 MPa were 6 × 10^−4^, 4 × 10^−4^, 3 × 10^−4^, and 3 × 10^−4^, respectively; the variations in the axial strain of the crack with the confining pressure were not obvious. During the stage of rapid crack propagation, the variations in the crack axial strain were 0.0354, 0.0984, 0.1768, and 0.2589, respectively; these variations of the crack axial strain in this stage increased obviously with the increase in the confining pressure. The variations in the above data indicated that the axial strain of the crack was less affected by the confining pressure in the stable expansion stage, which was due to the fact that this stage occurred before the yield stage when the coal body was mainly in a compressed state, the primary crack was compacted, and the new crack was in the germination and slow-expansion stage. In the stage of rapid crack propagation, the confining pressure had a great influence on the axial change in the crack. Because a large amount of energy accumulated before the failure of the coal body under a higher confining pressure, the new cracks developed and penetrated at a faster speed, so the amount of the change in the axial strain of the crack in the yield stage increased with the increase in the confining pressure.

## 5. Conclusions

A series of conventional triaxial compression tests and numerical simulations were carried out, the influence of the confining pressure on the deformation and failure of coal was analyzed, and then the crack-propagation characteristic parameters and damage variables were studied. The influence of the confining pressure on crack initiation and extension was analyzed, and the following conclusions were obtained:

(1) Based on the crack-propagation model, the variation in the axial strain of the initial propagation crack and the inelastic flexibility *S*_1_ of the crack with the confining pressure were obtained. The increase in the confining pressure had an effect of inhibiting the crack propagation; this inhibition effect was more obvious when the confining pressure range was 4 MPa–12 MPa. Therefore, the increase in the confining pressure showed a resistance effect on the deformation of the coal.

(2) The coal body changed from the compression state to the expansion state between the yield point and the peak point, and the increase in the confining pressure made the compression phenomenon of the coal body at the yield point and the expansion phenomenon at the peak point more obvious. The transition time of the two states became longer with the increase in the confining pressure, and the volume-expansion rate of the coal body became faster with the increase in the confining pressure after the expansion point, which indicated that the change in the confining pressure had a more significant effect on the damage to the coal body after the coal body entered the yield stage.

(3) The damage-accumulation process was related to the crack-propagation stage and confining pressure: in the stable crack-propagation stage, the damage accumulation began to accumulate gradually and the damage variable corresponding to the yield point decreased with the increase in the confining pressure. In the stage of rapid crack propagation, the confining pressure had a great influence on the axial change in the crack: the large amount of energy accumulated before the failure of the coal body under a higher confining pressure caused the new cracks to develop and penetrate at a faster speed, which resulted in a rapid increase in the damage variable.

## Figures and Tables

**Figure 2 ijerph-20-00105-f002:**
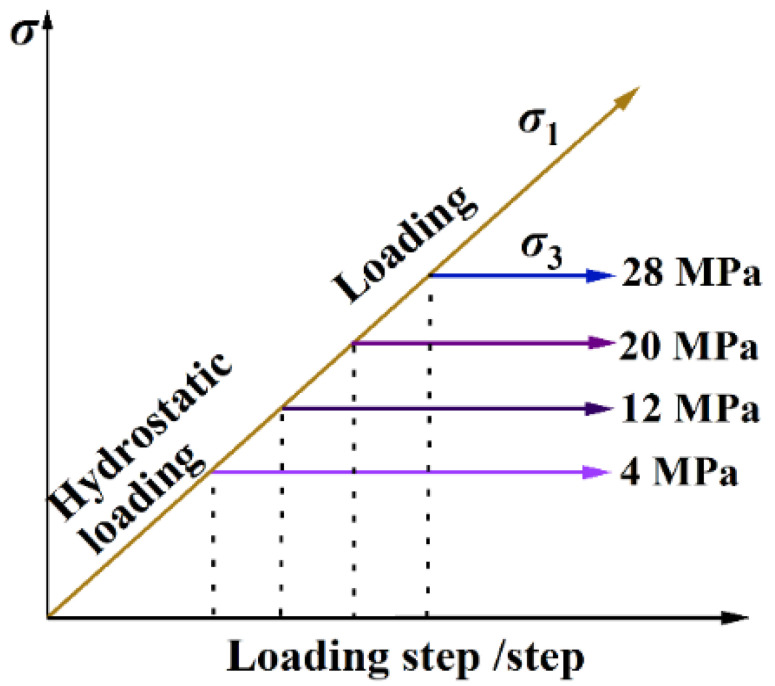
Stress-path diagram.

**Figure 4 ijerph-20-00105-f004:**
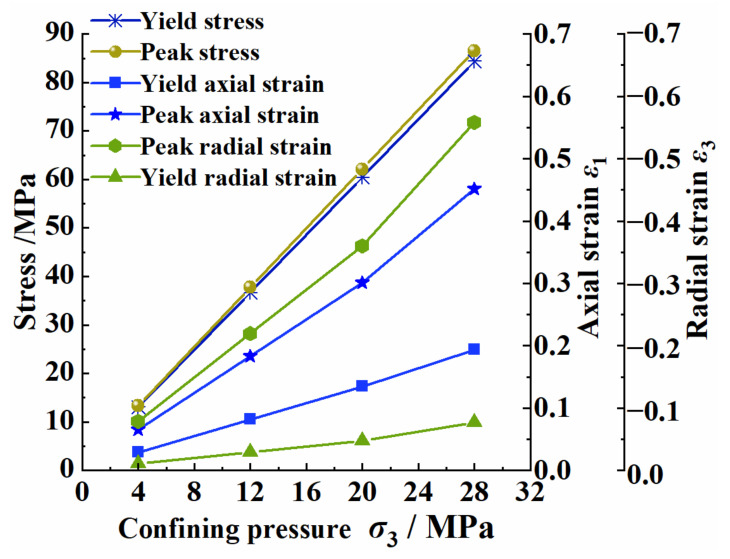
Stress and strain changes corresponding to yield and peak points under different confining pressures.

**Figure 5 ijerph-20-00105-f005:**
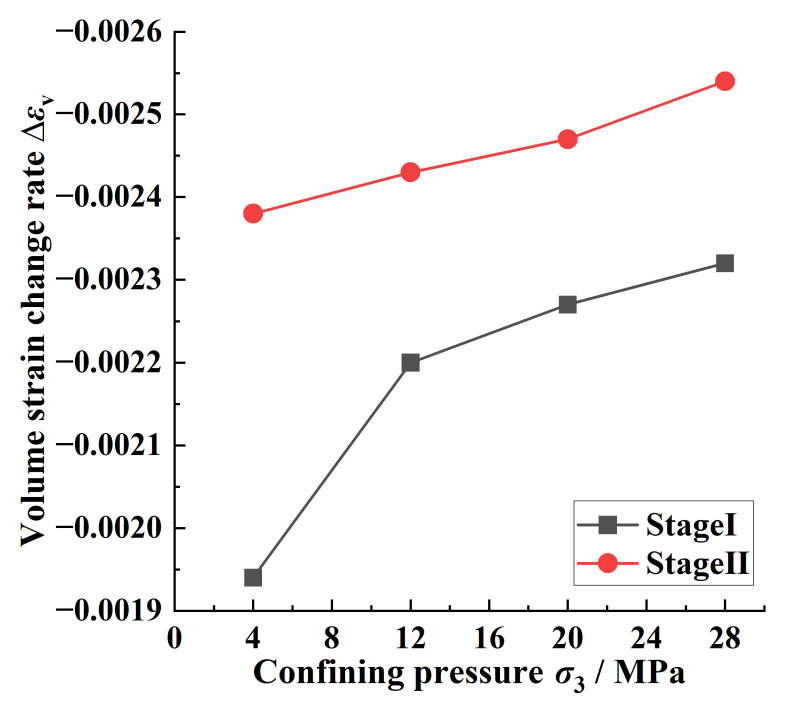
Variation rate of volumetric strain with confining pressure from yield point to peak point. Stage I is the stage from the yield point to the expansion point; Stage II is the stage from the expansion point to the peak point.

**Figure 6 ijerph-20-00105-f006:**
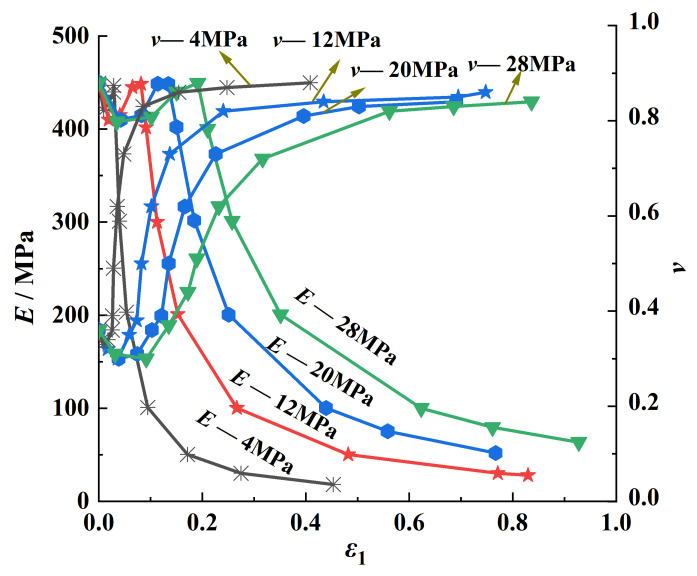
Variations in *E* and *v* with axial strain under different confining pressures.

**Figure 7 ijerph-20-00105-f007:**
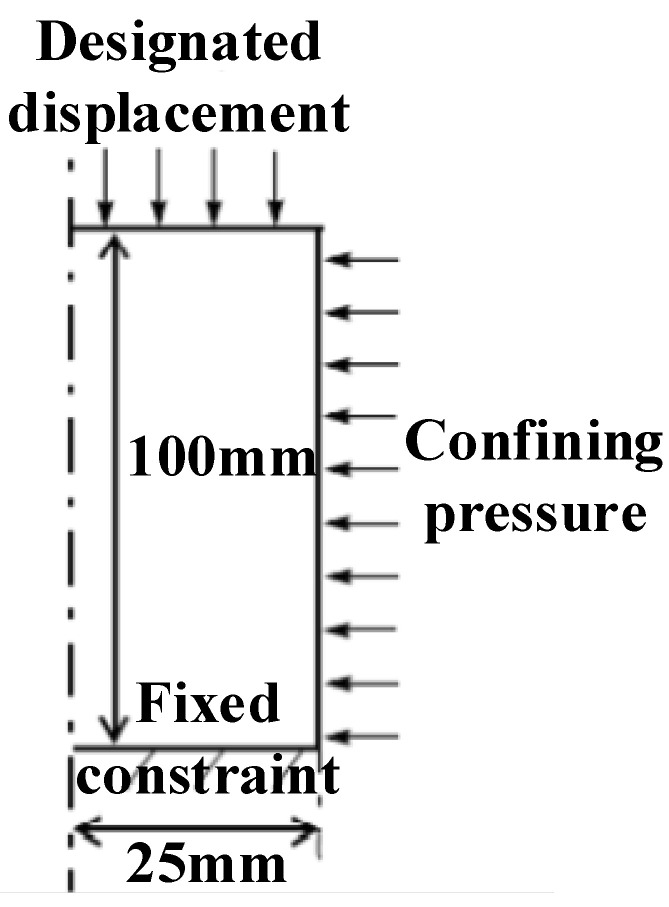
Boundary conditions of the model.

**Figure 8 ijerph-20-00105-f008:**
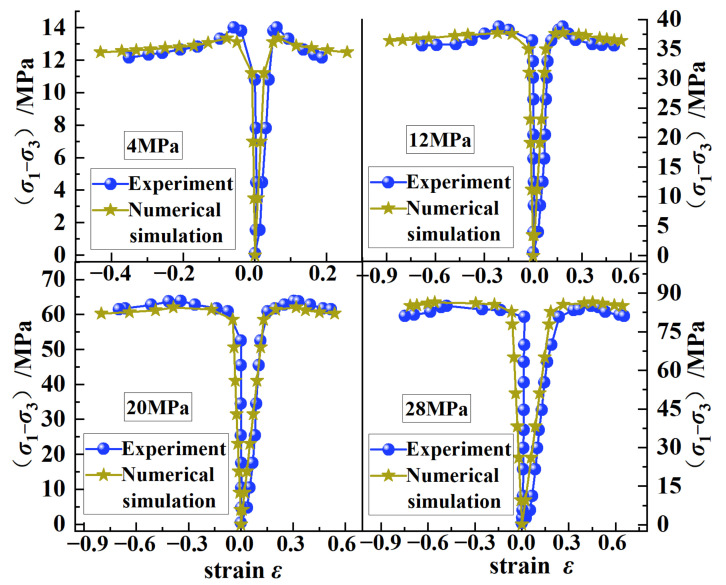
Stress–strain curves of laboratory test and simulation test.

**Figure 9 ijerph-20-00105-f009:**
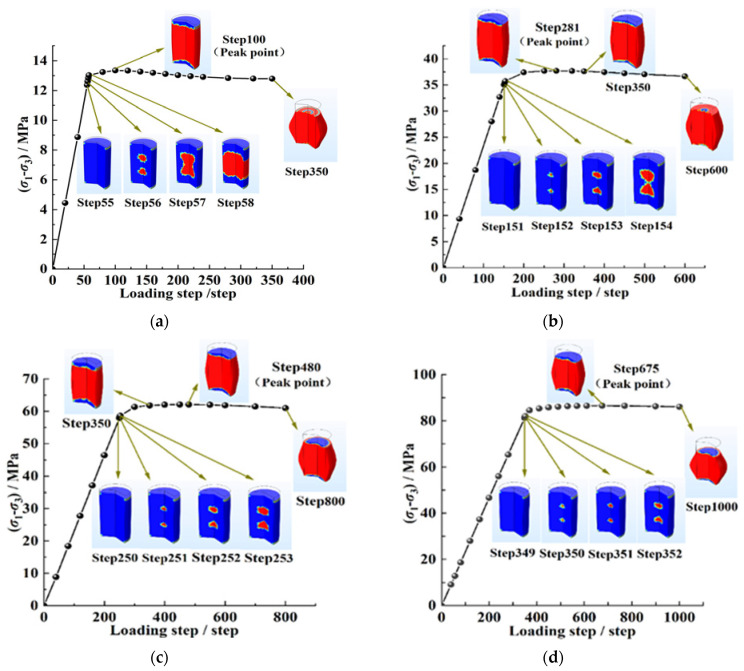
Plastic deformation of coal under different confining pressures varied with loading steps: (**a**) 4 MPa of confining pressure; (**b**) 12 MPa of confining pressure; (**c**) 20 MPa of confining pressure; (**d**) 28 MPa of confining pressure.

**Figure 10 ijerph-20-00105-f010:**
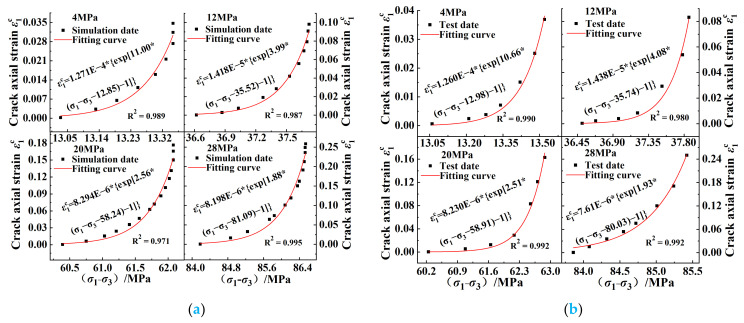
Verification of crack axial stress–strain−difference pre−peak curves under different confining pressures: (**a**) numerical simulation data validation; (**b**) laboratory data validation.

**Figure 11 ijerph-20-00105-f011:**
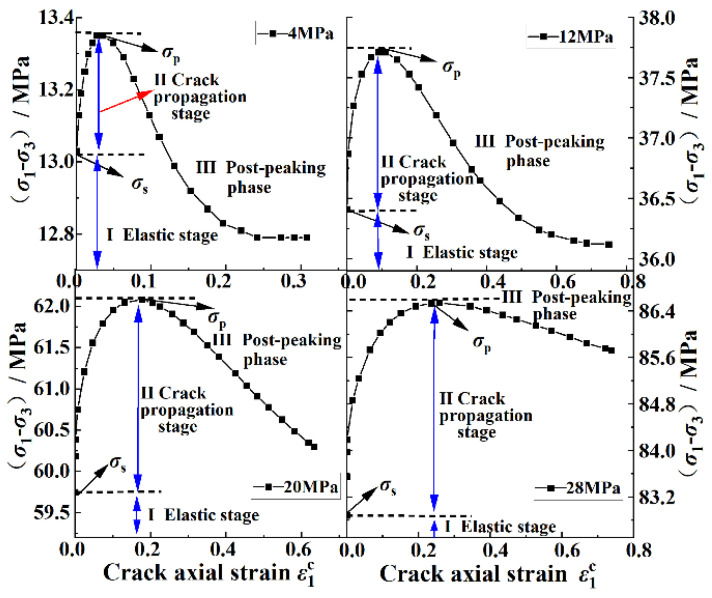
Axial stress–strain curves of cracks under different confining pressures.

**Figure 12 ijerph-20-00105-f012:**
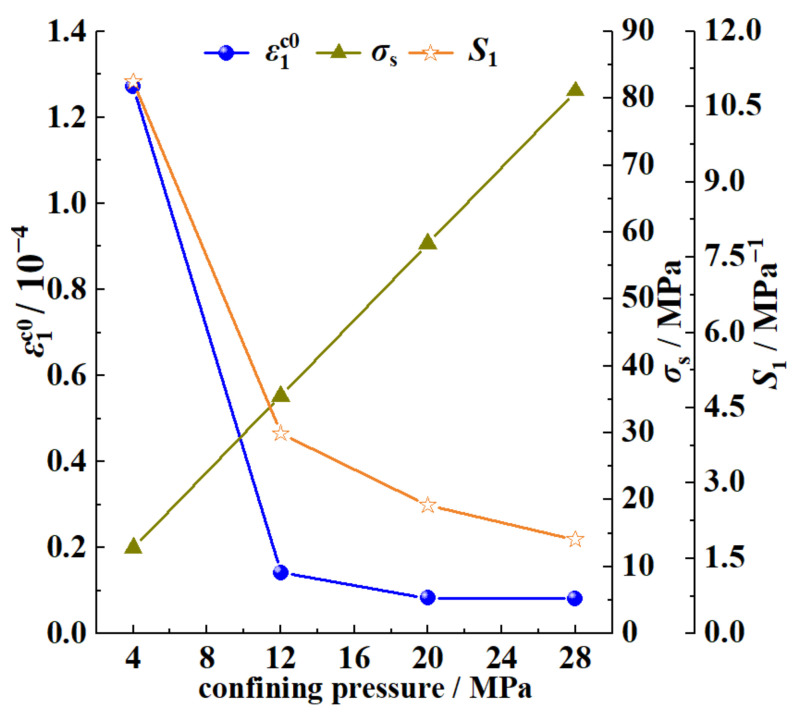
Curve of fitting parameters changing with confining pressure.

**Figure 13 ijerph-20-00105-f013:**
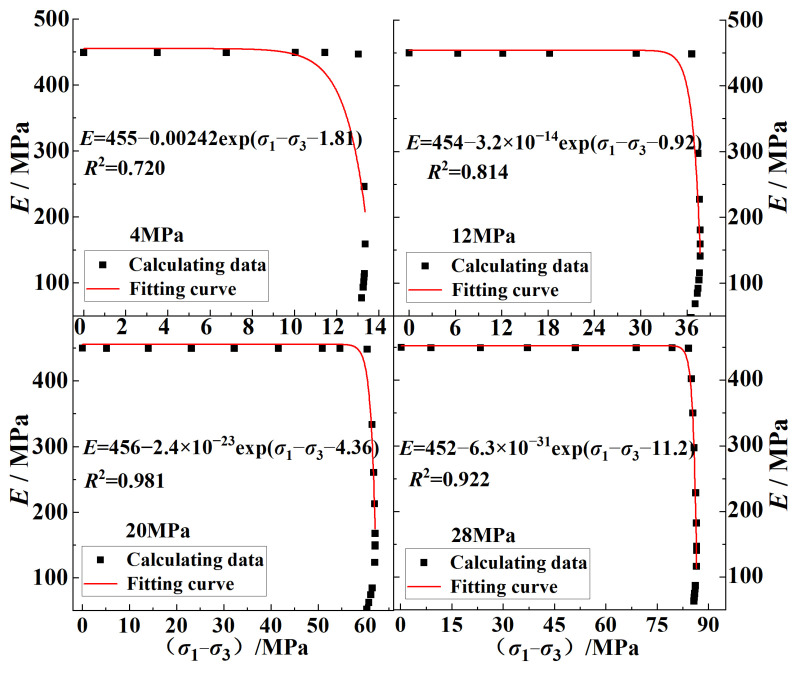
Fitting curves of deformation modulus with stress difference under different confining pressures.

**Figure 14 ijerph-20-00105-f014:**
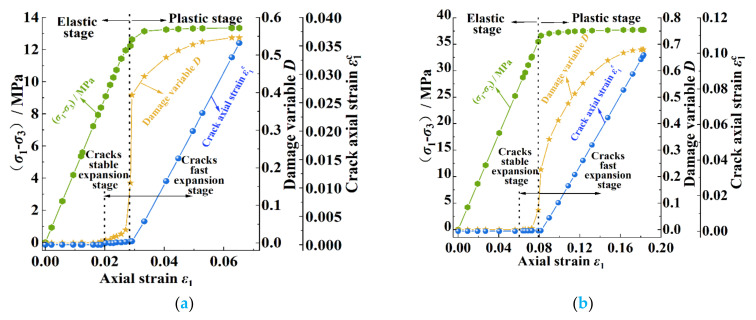
Cumulative curves of stress, strain, and damage under different confining pressures in pre-peak stage: (**a**) 4 MPa of confining pressure; (**b**) 12 MPa of confining pressure; (**c**) 20 MPa of confining pressure; (**d**) 28 MPa of confining pressure.

**Table 1 ijerph-20-00105-t001:** Model coal material parameters.

Parameter	Value	Unit	Parameter	Value	Unit
Elastic modulus (*E*)	450	MPa	Internal friction angle of coal (*φ*)	37	°
Density of coal (*ρ*)	1380	kg/m^3^	Coal expansion angle (*φ_r_*)	23	°
Poisson’s ratio of coal (*v*)	0.36	—	Coal body cohesion (*c*)	0.29	MPa

**Table 2 ijerph-20-00105-t002:** Fitting results of deformation modulus under different confining pressures.

Initial Confining Pressure/MPa	*a*	*k*	*b*	*D* _S_	*D* _C_
4	455	2.42 × 10^−3^	1.814	0.16	0.54
12	454	3.21 × 10^−14^	0.916	0.073	0.68
20	456	2.40 × 10^−23^	4.361	0.019	0.65
28	452	6.33 × 10^−31^	11.18	0.018	0.75

## Data Availability

Data are available upon reasonable request.

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
