# Peer review of "Influence of Confining Pressure on Nonlinear Failure Characteristics of Coal Subjected to Triaxial Compression"

_ijerph, 2022, doi:10.3390/ijerph20010105_

Round 1

Reviewer 1 Report

The manuscript entitled "The Influence of confining Pressure on the nonlinear failure characteristics of coal under triaxial compression" analyzes the influence of confining pressure on the nonlinear failure characteristics of coal by combining conventional laboratory tests with numerical simulation tests. This paper is innovative to some extent, but there are also some problems, which will be elaborated as follows:

1. The research status in the introduction is not very generalized, and the key points are not classified and summarized.

2. The test equipment in Figure 1 is recommended to be marked, at least the axial pressure and confining pressure loading parts described.

3.Page 3, The test results in Fig. 3 are very coarse, and the deformation stage of the sample is not analyzed in stages according to the data map.

4. Page 4,What are the definitions of yield and peak points?

5. Section 3.3,“with the increase of confining pressure, the time of plastic deformation of coal body becomes longer, which was not verified in physical experiments.

6. Overall, the analysis of physical experiment results is relatively crude. In addition, the test results are not fully combined with physical experiments and numerical simulations.

Reviewer 2 Report

This paper studied the influence of confining pressure on nonlinear failure characteristics of coal subjected to triaxial compression. The crack propagation model was used. The results showed that the increasing confining pressure could inhibit crack propagation. Moreover, after the coal body enters the yield stage, the change of confining pressure has a more significant influence on the damage of coal body. I have the following comments for the authors to consider.

(1) The English writing can be improved. Some sentences can be shorten. For example, "in order to" can be replaced with "to".

(2) In the introduction, the authors mentioned that "Fractures in natural rocks play an important role in determining the strength, deformability and failure behavior of rock masses". In fact, to increase the strength of fractured rock masses, rock reinforcement methods are always used in underground mining. This should be indicated.

(3) In the introduction, "Aiming at the influence of confining pressure on the deformation" should be replaced with "Aiming at revealing the influence of confining pressure on the deformation".

(4) For the experimental triaxial test, the authors used the confining stress of 4MPa, 12 MPa, 20 MPa and 28 MPa. I recommend that the author briefly explain why those four values were used.

(5) For the numerical simulation, which software did the authors use?

(6) For the values used in Table 1, can authors briefly explain how the coal body cohesion is obtained?

(7) Which constitutive model was used for this numerical specimen?

Round 2

Reviewer 2 Report

Publish.